**Data Availability Statement:** All relevant data are within the manuscript and its Supporting Information files.

# Factors affecting enrollment status of households for community based health insurance in a resource-limited peripheral area in Southern Ethiopia. Mixed method

**Mustefa Glagn Abdilwohab**👁️*, **Zeleke Hailemariam Abebo, Wanzahun Godana, Dessalegn Ajema, Manaye Yihune, Hadiya Hassen**

Department of Public Health, College of Medicine and Health Sciences, Arba Minch University, Arba Minch Town, Ethiopia

* mustesami02@gmail.com

## Abstract

### Background

Despite the efforts made by the government of Ethiopia, the community-based health insurance (CBHI) enrollment rate failed to reach the potential beneficiaries. Therefore, this study aimed to assess the enrollment status of households for community-based health insurance and associated factors in peripheral areas of Southern Ethiopia.

### Methods

We conducted a community based cross-sectional study design with both quantitative and qualitative methods. Systematic random sampling was employed to select 820 households from 27, April to 12 June 2018. A pretested structured questionnaire, in-depth interview, and focus group discussion guiding tool were used to obtain information. A binary logistic regression model was used to assess the association between independent and outcome variables. A P-Value of less than 0.05 was taken as a cutoff to declare association in multivariable analysis. Qualitative data were analyzed manually using the thematic analysis method.

### Results

Out of 820 households, 273[33.30%; 95% CI: 29.9–36.20] were enrolled in the community based health insurance scheme. Having good knowledge [AOR = 13.97, 95%CI: 8.64, 22.60], having family size of greater than five [AOR = 1.88, 95% CI: 1.15, 3.06], presence of frequently ill individual [AOR = 3.90, 95% CI: 2.03, 7.51] and presence of chronic illness [AOR = 3.64, 95% CI: 1.67, 7.79] were positively associated with CBHI enrollment. In addition, poor quality of care, lack of managerial commitment, lack of trust and transparency, unavailability of basic logistics and supplies were also barriers for CBHI enrollment.

**Funding:** Arba Minch University is funding this research work with a project grant code of GOV/ AMU/TH.3.1/CMHS/HO/02/10. The website of the university is http://www.amu.edu.et/. The funders had no role in study design, data collection and analysis, decision to publish, or preparation of the manuscript.

**Competing interests:** The Author declare that there is no conflict of interest.

## Conclusion and recommendation

The study found that lower community based health insurance enrollment status. A higher probability of CBHI enrollment among higher health care demanding population groups was observed. Poor perceived quality of health care, poor managerial support and lack of trust were found to be barriers for non-enrollment. Therefore, wide-range awareness creation strategies should be used to address adverse selection and poor knowledge. In addition, trust should be built among communities through transparent management. Furthermore, the quality of care being given in public health facilities should be improved to encourage the community to be enrolled in CBHI.

## Introduction

Primary health care financing is the structural aspects of health systems that play an essential role in ensuring universal health coverage (UHC) [1]. It includes three interrelated functions: mobilization and collection of funds, pooling of prepaid funds and allocation of resources including purchasing and paying for services [2]. The goal of UHC is to eliminate the financial difficulty associated with obtaining the necessary health services that ensure the wellbeing and productivity of a society. Mechanisms that offer health security through risk pooling like a Community Based Health Insurance Scheme is one of the possible tools in achieving UHC [1].

Community Based Health Insurance (CBHI) is part of the Ethiopian government broader health care financing reform strategy which aims to promote equitable access to health care, increase financial protection, promote cost sharing between the government and citizens, and enhance domestic resource mobilization for the health sector and social inclusion in health [3]. Based on the concepts of mutual aid and social solidarity, CBHI is designed mainly for people who live and work in the rural areas or urban informal sectors that are unable to get public, private, or employer-sponsored health insurance. It is an alternative financing mechanism through contributing some amount of money that is owned, designed, and managed by the members [4–6]. It is also supposed to reduce unforeseeable or unaffordable healthcare costs (in the case of illnesses) to regularly paid premiums [7].

After considering CBHI significance and sharing experience from forefront countries, in June 2011, with the aim of enhancing access to health care and reducing the burden of out-of-pocket health care expenditure, the government of Ethiopia rolled out a pilot CBHI scheme. The pilot scheme tailored to rural households and urban informal sector workers in 13 districts situated in four main regions [Tigray, Amhara, Oromia and Southern Nation Nationalities and Peoples' Region] of the country. After three years of piloting, the government decided to expand community-based health insurance schemes to 161 districts of the country [8].

The overall enrollment rate of households for CBHI in the pilot districts of Ethiopia in 2013 reached 52.4% [9]. However, only 1.2% of Ethiopian citizens had health insurance from either private or public agencies [10]. Despite having over a decade of experience in the sub Saharan region, community-based health insurance is not prevalent in Eastern African countries. Tanzania, Uganda, and Kenya are countries in East Africa that have experience with community-based health insurance though their coverage has not reached more than 15% [11]. However, a high rate of coverage is seen in Rwanda where coverage has scaled up from 35% in 2006 to 85% in 2008 [12].

Evidence from meta-analysis suggested that CBHI enrollment was positively associated with household income; educational status, age, marital status, and sex of the head of the household; family size; trust in the scheme management and presence of chronic illness episodes in the household. However, there are controversial findings like the presence of acute illness episodes and the presence of elderly persons in the household had a negative association with enrollments in CBHI [13]. Generally, the existing health insurance coverage in the Sub-Saharan region is very low and the enrollment rate is affected by multiple factors [10].

Even though Ethiopia has been implementing the CBHI scheme since 2011 to promote the health of poor rural and urban informal residents, the enrollment rate is still very low when compared to the potential beneficiaries. A need for further studies on community-based health insurance enrollment was recommended by the Ethiopian health insurance agency CBHI evaluation team [8]. Specific to our study area the socio-cultural context is very different from those in the pilot district and our target districts were not included in the pilot study. In addition, the factors affecting CBHI enrollment are not well described in Ethiopia in general and in the study setting in particular. Therefore, the study aimed to assess factors affecting the enrollment status of households for community-based health insurance in a resource-limited peripheral area in Southern Ethiopia. A better understanding of the factors and CBHI enrollment status could be helpful for policy and program interventions and to direct resources most effectively and efficiently.

## Methods and materials

### Study design, setting, and population

A community-based cross-sectional study (with both quantitative and qualitative methods) was conducted among households living in CBHI implemented districts of Segen area and South Omo zones, Southern Ethiopia from 27, April to 12 June 2018. Household heads and/or spouses who were working in the formal sectors were excluded. Both Segen area and South Omo zones are located in the resource-limited peripheral zones of Southern region. There are eight districts in South Omo zone; among them, four of the districts were implementing community-based health insurance during the study period. Among the districts which were implementing CBHI, two of districts were selected for this study (South Ari and Benatsemay). In addition, Derashe district was selected from Segen area zone because it was the only district in Segen area zone implementing CBHI during the study period.

### Sample size and sampling technique

The sample size for this study was determined using single population proportion formula, taking into assumptions: 95% confidence level, 5% margin of error, and 52.4% proportion of households enrolled in community-based health insurance on pilot CBHI implemented districts in Ethiopia [9] and using design effect of two and 10% non-response rate. The final sample size for the number of households under the study was 842. A multi-stage sampling technique was employed to reach the study participants.

First, fifty percent (50%) of the districts within the South Omo Zone were selected using the lottery method. And one district currently implementing CBHI from Segen area zone was purposively included. Secondly, 25% of kebeles (the smallest administrative unit in Ethiopia) with in the selected districts were included in the study using computer generated simple random sampling technique. Lastly, households were selected using systematic random sampling technique after proportionally allocated to the kebele based on size.

## Operational definitions

### CBHI enrolled households

Households who claim to be registered for CBHI and who had membership card at hand during the data collection period were considered as enrolled, otherwise not enrolled.

Participants' **Knowledge about CBHI** was assessed using five questions that consisted of concepts, roles and beneficiaries' of community based health insurance. Participants correctly responded to this questions were categorized as 'correct response' otherwise 'not correct response'. Each item was equally weighted. Thus, each correct response had a score of 1 and each wrong response had a score of 0. Hence, the aggregate score for all knowledge questions would range from 0–5 points. Participants' overall knowledge was categorized as **good** if the score was 4 and 5 ($> = 70\%$) points and otherwise poor.

### Data collection tools and procedures

**Quantitative data collection.**    The pretested evaluation tool from the federal democratic republic of Ethiopia health insurance agency was adopted for this study [8]. Interviewer administered structured and pretested Amharic version questionnaire were used to collect data. The respondents were the head of the households. The data on household wealth index was collected by asking ownership of selected assets based on Ethiopian demographic and health surveillance (EDHS) 2016 wealth index variables [14].

**Qualitative data collection.**    Focus Group Discussion (FGD) and in-depth interview guide were used to collect the qualitative data. The data collectors were masters of public health holders with experience in qualitative data collection and fluent speakers of the local language. Key informants' interview was held for three district health office heads, three district CBHI coordinators, and 9 health extension workers (HEWs). The key informant interviews were held at the offices, each interview lasting 30–35 minutes. The interviews were audio-recorded. Two FGDs in each district, one among CBHI members and another among nonmembers were conducted. A total of six FGDs (with a group of 7–11 participants) were conducted. The FGD lasted for 80–100 minutes and the discussions were audio-recorded. Participants were engaged in informal conversations in the form of unstructured spontaneous discussions to get the opportunity to ask pertinent questions on different occasions. This could minimize the possibility of participants altering their response purposefully or holding back information on sensitive issues such as disclosure of any attempt to abuse or fraud of the collected money, but that is important to our study.

**Data quality control.**    A pretest was done prior to actual data collection by recruiting 42 households out of the study setting. The final version of the questionnaire was translated into Amharic language and again translated back to English to check the consistency.

The data collectors and supervisors were given two days of intensive training on the overall data collection procedure, ethical issues, and the purpose of the study. After the pretest relevant modifications were made before the commencement of the actual data collection. Supervisors have checked the collected data for completeness and consistency throughout the data collection period. The investigators trained the moderators who were familiar with the local language, to conduct, observe, and record the FGDs and in-depth interviews. One moderator facilitated the discussion, whereas the other concentrated on note-taking and audio recording.

**Quantitative data analysis.**    After checking and correcting errors, data were entered into Epi -data version 3.1 then data were exported to Statistical Package for Social Science (SPSS) version 21software for further analysis. Descriptive analysis of data was indicated using numerical summary measures. Outliers were also checked. The level of

analysis for this study was the household, considering that enrollment in Ethiopian CBHI is currently at the household level. The outcome variable was treated as a binary outcome (1 for enrolled and 0 for not enrolled households). Binary logistic regression was carried out to assess the association of different independent variables with the dependent variable after assumptions of logistic regression were checked. Independent variables having P≤0.25 on bivariate binary logistic regression analysis were considered as candidates for the multivariable logistic regression analysis. The final model was fitted using backward conditional variable selection methods and Hosmer and Lemeshow's test of model adequacy was 0.90. Multivariable logistic regression analysis was carried out to identify factors having statistically significant associations with the dependent variable. Finally, the adjusted odds ratio AOR with a 95% confidence interval, and p-value < 0.05 was used to determine a significant association between CBHI enrollment status and the independent variables.

**Principal component analysis.** This study was conducted in rural setting in which households cannot clearly define their wealth status. Therefore, in our study principal component analysis (PCA) was used to calculate the composite wealth index. Initially, 25 items composed of different assets were entered in the analysis. If a variable/asset was owned by more than 95% or less than 5% of the sample, it was excluded from the analysis because it would not help to distinguish between higher and lower economic status of households. We have checked assumptions of PCA using Kaiser-Meyer-Olkin measure of sampling adequacy (> 0.5). In each step, variables with anti-image correlations and communalities less than 0.5, having a loading (correlations higher than 0.4) in more than one component (having complex structure), and a single variable loading in a component were removed until the iterations fulfilled the inclusion criteria. Finally, two components which explained a total variance of 63.4% were extracted from the PCA. A factor score of this component was used to categorize the household wealth index into lowest, second, middle, fourth, and highest wealth quantile.

**Qualitative data analysis.** Data were collected, transcribed, quoted, coded, and analyzed manually using the thematic analysis method. Themes and categories emerged from the text data through repeated reading. The FGD and in-depth interview guide and transcripts were designed in line with barriers to CBHI enrolment. Based on this theme, a coding system was developed that represented common topics encountered in the transcript review. Codes were refined throughout the data analysis period. From this process, descriptive categories were developed to show the factors which were then characterized with a "name" to describe a basis of explanation for the observed phenomenon. Finally, qualitative data results were presented with the quantitative result through triangulation. Quotes that were most useful in explaining the quantitative findings were selected and written under each quantitative finding to strengthen the quantitative findings. Quotes that were not demand-side factors are written separately as supply-side factors and presented based on their theme.

## Ethical considerations

Ethical approval was obtained from the Institutional Ethics Review Board [IRB] of Arba Minch University. Official permission letter was obtained from both Segen area and South Omo zonal health department and the data collection began after permission and cooperation letter was written to all three respective districts and respective kebele (the smallest administrative unit in Ethiopia) where the study was carried out. Household head informed written consent was obtained and the respondents were assured of confidentiality.

## Results

### Socio-demographic and economic characteristics of the study participants

A total of 842 participants/household heads were recruited, of whom 820 (97.38%) consented to participate in the present study. The mean age of the participants was 40.0 ± 11.05 years (SD) with an age range between 20 and 76 years. Out of the study participants, 43.7% had no formal education and 62.4% of them had a family size of less than or equal to five. "**Table 1**"

### Medical-related factor

Out of the study participants, 127(15.5%) reported that one or more of the household members had a chronic non-communicable disease. Of them, 84.25% of households were enrolled in a community-based health insurance scheme. One hundred forty-six (17.8%) of the study

**Table 1. Socio-demographic and economic characteristics of the study participants in Segen area and South Omo Zones, Southern Ethiopia, 27 April to 12 June 2018 (n = 820).**

| Variables | Category | frequency | Percent (%) |
|---|---|---|---|
| **Age of head of the household** | < = 25 years | 58 | 7.1 |
| | 26–34 years | 225 | 27.4 |
| | 35–44 years | 250 | 30.5 |
| | 45–54 years | 186 | 22.7 |
| | Above 55 years | 101 | 12.3 |
| **Sex of the participant (household head)** | Female | 147 | 17.9 |
| | Male | 673 | 82.1 |
| **Educational status of the head of the household** | No formal education | 358 | 43.7 |
| | Primary school | 344 | 42.0 |
| | Secondary school | 55 | 6.7 |
| | Above Secondary | 63 | 7.6 |
| **Marital status of head of the household** | Single | 41 | 5.0 |
| | Married | 665 | 81.0 |
| | Divorced | 39 | 4.8 |
| | Widowed | 58 | 7.1 |
| | separated | 17 | 2.1 |
| **Occupational status of head of the household** | Farmer | 468 | 57.1 |
| | pastoralist | 115 | 14.0 |
| | Merchant | 161 | 19.6 |
| | Daily laborer | 40 | 4.9 |
| | other | 36 | 4.4 |
| **Family size** | = < 5 | 512 | 62.4 |
| | >5 | 308 | 37.6 |
| **Wealth quantile** | Lowest | 133 | 16.2 |
| | Second | 104 | 12.7 |
| | Middle | 402 | 49.1 |
| | Fourth | 20 | 2.4 |
| | Highest | 161 | 19.6 |
| **The presence of children age less than 18 years in the household** | yes | 497 | 60.6 |
| | No | 323 | 39.4 |
| **The Presence of elderly people (65+years)in the household** | Yes | 179 | 21.8 |
| | No | 641 | 78.2 |

participants reported that there was an episode of illness due to communicable disease among one or more members. Among the households who encountered an episode of communicable disease 82.8% of them were enrolled in community-based health insurance.

## Physical accessibility of health facilities

Regarding the time taken to reach the nearest health facilities, 37.3%, 59%, and 3.7% of the participants reported that they spent less than or equal to one hour, one to two hours, and greater than or equal to two hours to reach the facilities and receiving health care, respectively.

## Knowledge of the study participants regarding CBHI

A total of 561 (68.4%) of the respondents have heard about community-based health insurance messages. Among the participants who have heard about CBHI messages, 47.4% were enrolled in the scheme; and their main source of sensitization information was public meetings followed by neighbors/friends (**Fig 1**).

The participants reported that there were no adequate community sensitizations on the meaning and benefit packages of the CBHI scheme by the stakeholders who own the lead.

*"Although we heard the name, community-based health insurance from our village, nobody from government or any other concerned bodies told us its importance, and even it's meaning." (FGD- CBHI nonmembers)*

A total of 446 (54.4%) of the study participants had poor knowledge related to community-based health insurance and the remaining 45.6% of the participants had good knowledge related to CBHI. "**Table 2**"

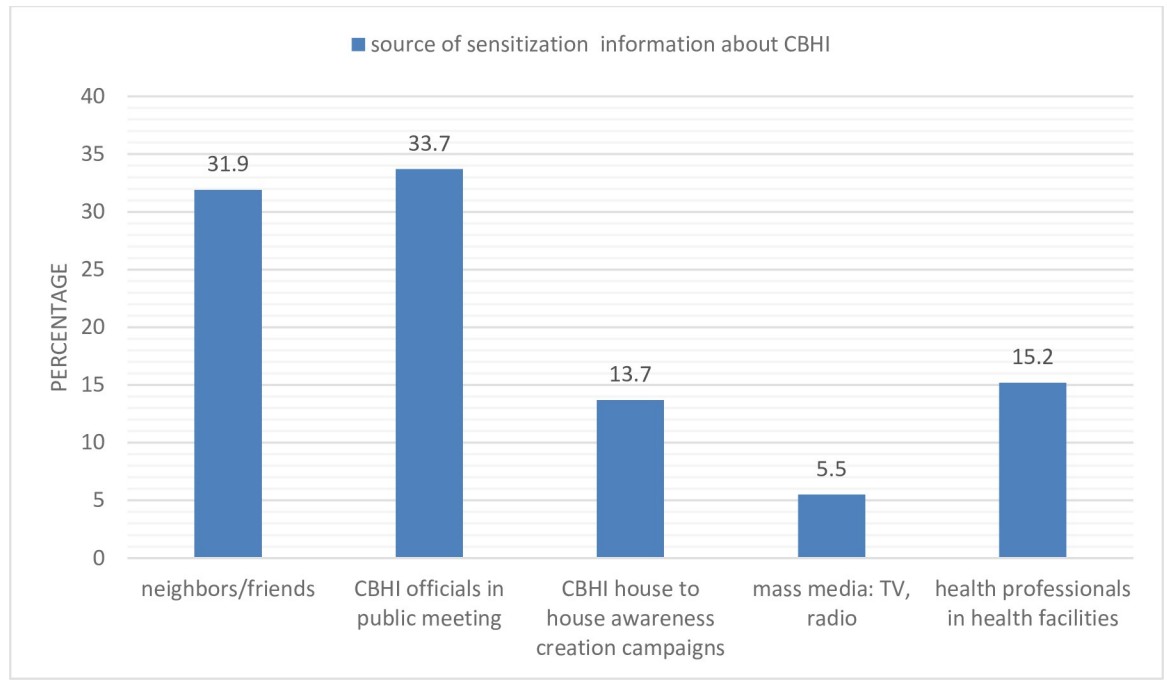

**Fig 1. Source of information regarding CBHI in Segen area and South Omo Zones, Southern Ethiopia, 27 April to 12 June, 2018.**

**Table 2. Knowledge of the study participants related to CBHI in Segen area and South Omo zones, Southern, Ethiopia 27 April to 12 June 2018.**

| variables | Correctly responded | Not correctly responded |
|---|---|---|
| Only the very poor who cannot afford to pay for healthcare needs to join the schemes | 400 (48.8%) | 420 (51.2% |
| Under the CBHI program, you pay money (premiums) in order for the CBHI to finance your future health care needs | 489 (59.6%) | 331(40.4%) |
| CBHI program is like a savings scheme, you will receive interest and get your money back | 412 (50.2%) | 408 (49.8%) |
| If you do not make claims through CBHI, your premium will be returned | 391 (47.7%) | 429(52.3%) |
| Only those who fall sick should consider enrolment in CBHI | 440(53.7%) | 380 (46.3%) |

## Community-Based Health Insurance (CBHI) enrollment status

Out of the study participants 273[33.3%; 95% CI: 29.9–36.2] were enrolled in the CBHI scheme, A total of 289 (35.2%) of the respondent reported that at least one of the member of the family participated on any community-based health insurance-related meeting/training.

## Reasons not to enroll in community-based health insurance

Inadequate information on CBHI, limited availability of health services, poor quality of health care provided in public health facilities and illness do not frequently occur in their homes etc. were among reasons for non-enrolment to CBHI (**Fig 2**).

## Perceived quality of care

Some participants argued that the perceived quality of service provided at public facilities is poor. Likewise, they also complained unavailability of logistics and supplies including drugs. Furthermore, they want to see how the previously enrolled CBHI members benefited from the scheme. *"The care given to us at the hospital/health center is poor; when we go to the hospital or health center we weren't getting medication even for headache. . . .health professionals prescribe to the private drug stores (pharmacies). . . .in this situation, how can we get interested to be CBHI members? Even we weren't getting quality service when we pay out of our pocket. We don't think, we can get adequate service for free. . . .let's see how the service provision will be changed for those who previously enrolled . . . .after that we might also enroll for membership (FGD discussant, CBHI non-members).*

## Affordability of the premium, expectations, and experience from the CBHI program

Among the respondents, 279 (34%) of them were enrolled in other forms of active local social solidarity groups (e.g. idir (savings for funeral ceremony), equb (is a traditional means of saving in Ethiopia and exists completely outside the formal financial system/ "credit union"), microfinance, and other informal systems etc.) in their respective areas. Two hundred seventy-nine (34.0%) of the respondents agreed that the time of collection of the regular premium is convenient for their households. Whereas, 93.2% and 93% of the participants reported that both registration fees and regular premium are easily affordable respectively.

*One of the female FGD participants said "the premium we have contributed is reasonable—if you go to a private clinic, for a single visit you may pay 1000 birr or more. . . .when you*

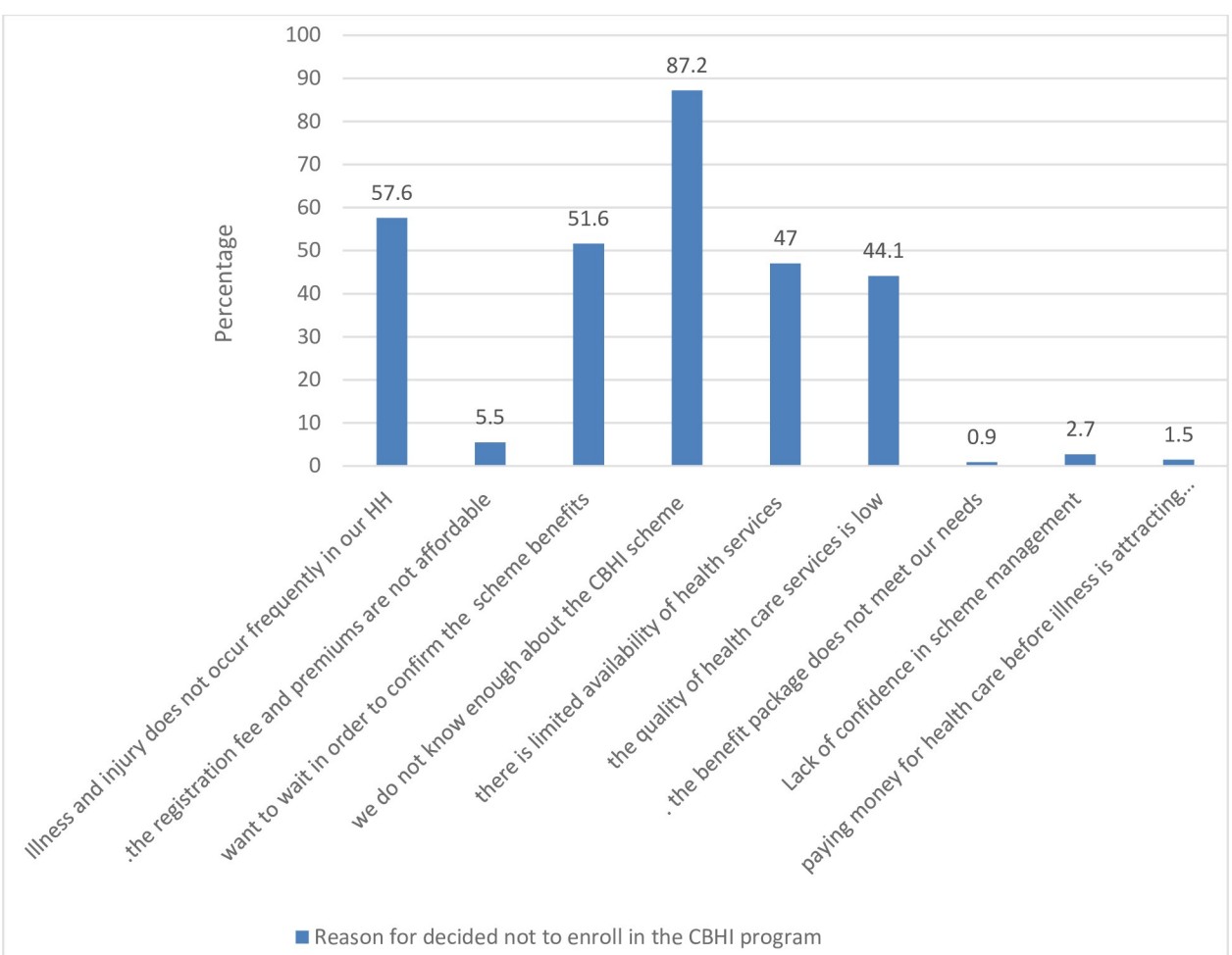

**Fig 2. The study participants reason for decided not to enroll in CBHI in Segen area and South Omo Zones, 27 April to 12 June, 2018 (n = 547).**

*compare the premium payment for CBHI with what the private clinics charge us for their service it's negligible.. ..it is a gift from the government to the poor (who cannot afford to pay) for medical expenditure. . ...the problem is the service has not started yet.* "**Table 3**"

## Trust management

People's trust in CBHI management is a facilitator for insurance enrollment decisions. However, the discussant raised trust issues on CBHI management teams because of their past

**Table 3. Community based health insurance experience of the enrolled households in Segen area and South Omo Zones, 27 April to 12 June 2018 (n = 273).**

| variables | Response | | |
|---|---|---|---|
| | Agree | Indifferent | Disagree |
| The local CBHI agent tries hard to solve CBHI implementation problem | 164(60.1%) | 3(1.1%) | 106(38.8%) |
| The community (CBHI members) has the right to guide and supervise the activities of the CBHI management | 89(32.6%) | 42(15.4%) | 142(52.0%) |
| The local CBHI management is trustworthy | 104 (38.1%) | 58 (21.2%) | 111 (40.7%) |
| I am satisfied with the experience at the local CBHI office when I go to register? | 36 (13.2%) | 13 (4.8%) | 224 (82.0%) |
| I am satisfied with the local CBHI office when I go to pay the regular contribution (premium) | 58 (21.2%) | 25 (9.2%) | 190 (69.6%) |

experience in the contributed money for different purposes.: *". . .Because lower-level managers had a previous history of fraud and corruption on public resources, currently, we have no trust in them. . . .now the government is sending them to join higher education without any punitive measures. . ..in addition, we did not see the fruit of previously contributed money for different purposes. . .honestly speaking, we don't trust them". (FGD- CBHI member & non—members)*

### Community involvement

Poor community involvements were reported; i.e., community leaders, religious leaders, elders, and others in addition to existing government structures can play a crucial role at woreda and kebele levels in sensitization and awareness-creation activities that can facilitate the acceptance of the schemes and increased enrollment rates.

*"We have no idea how much birr is collected, how many households were enrolled. . .where the premium is pooled. . .. simply they had taken our money but the service is not started yet. . .we need to get our money back otherwise the service should be started." (FGD-CBHI member)*

### Factors associated with CBHI enrollment

Educational status, sex, age, marital and occupational status of the household head, family size, presence of children whose age are ($\leq$18 years) & elders (65+ years), presence of a person with chronic disease (NCDs) & frequently ill individual due to a communicable disease, knowledge, and distance from the health facility were eligible for multivariable logistic regression.

In multivariable binary logistic regression knowledge, family size, presence of children whose age are ($\leq$18 years) & elders (65+ years), presence of a person with a chronic non-communicable disease (NCDs) & frequently ill individual due to communicable disease and educational status of the household head remained to have an association with CBHI enrollment. The odds of enrolling in CBHI among families who had a family member of greater than five was about 2 times higher than a family member of five or less (AOR = 1.88,95% CI: 1.15, 3.06).

The odds of enrolling in CBHI among households who had children whose age is ($\leq$18 years) was 3.64 times higher than families with no children under 18 years (AOR = 3.64, 95% CI: 2.09, 6.33). The odds of enrolling in CBHI among households with the presence of elders were 2.60 times higher than without elders (AOR = 2.60, 95%CI: 1.45, 4.65). Apart from this the odds of enrolling in CBHI among respondents who had good knowledge were around 14 times higher than those with poor knowledge (AOR = 13.97, 95%CI: 8.64, 22.60). The odds of enrolling in CBHI among households with a presence of a person with chronic disease in the household were 3.64 times higher than households with no chronic disease (AOR = 3.64, 95% CI: 1.67, 7.79). The odds of enrolling in CBHI among households head who had primary education were 3 times higher than those with no formal education (AOR = 3.06, 95% CI: 1.88, 4.99). Similarly, the odds of enrolling in CBHI among frequently ill individual due to communicable disease in the household was around 4 times higher than those with no frequently ill person in the household(AOR = 3.90,95%CI:2.03,7.51)."**Table 4**"

### Supply-side factors

**Lack of managerial commitment.**   A higher tendency of assuming the CBHI related mobilization is the role of health extension workers only by district managers was reported. The weak leadership and commitment of managers at district level are affecting the progress of CBHI scheme enrollment and potential beneficiaries. *". . .the managers think CBHI*

**Table 4. Multivariable logistic regression analysis to identify associated factors of CBHI enrollment among households in Segen area and South Omo zones, 27 April to 12 June 2018 (n = 820).**

| Variables | Category | Enrolled in CBHI | Not enrolled in CBHI | COR (95% CI) | AOR(95% CI) | P-value |
|---|---|---|---|---|---|---|
| **Family size** | < = 5 | 86 (16.8%) | 426(83.2%) | 1.00 | 1.00 | |
| | >5 | 187(60.7%) | 121(39.3%) | 7.65(5.52,10.60) | 1.88(1.15, 3.06) | 0.012 |
| **Children whose age are** (≤18 years) are present in the HH | yes | 243(48.9) | 254(51.1%) | 9.34(6.17,14.14) | 3.64(2.09, 6.33) | .001 |
| | No | 30(9.3%) | 293(90.7%) | 1.00 | 1.00 | |
| **Elderly people in the household (65+ years) are present in the HH** | yes | 128(71.5%) | 51(28.5%) | 8.58(5.90,12.47) | 2.60(1.45, 4.65) | .001 |
| | No | 145(22.6%) | 496(77.4%) | 1.00 | 1.00 | |
| **A person with chronic disease in the household** | yes | 107(84.3%) | 20(15.7%) | 16.98 (10.21,28.2) | 3.64(1.67, 7.79) | .001 |
| | No | 166(24.0%) | 527(76.0%) | 1.00 | 1.00 | |
| **Frequently ill individual due to communicable disease in the household** | Yes | 121(82.9%) | 25(17.1%) | 16.62 (10.42,26.5) | 3.90(2.03, 7.51) | .001 |
| | No | 152(22.6%) | 522(77.4%) | 1.00 | 1.00 | |
| **Knowledge** | good | 240(64.2%) | 134(35.8) | 22.41 (14.83,33.8) | 13.97 (8.64,22.6) | .001 |
| | poor | 33(7.4%) | 413(92.6%) | 1.00 | 1.00 | |
| **Educational status of the household head** | No formal education | 87(24.3%) | 271(75.7%) | 1.00 | 1.00 | |
| | primary | 153(44.5%) | 191(55.5%) | 2.45(1.81,3.44) | 3.06(1.88, 4.99) | .001 |
| | secondary | 16(29.1%) | 39(70.9%) | 1.27(0.07,2.39) | .98(.37, 2.59) | .981 |
| | Above secondary | 17(27.0%) | 46(73.0%) | 1.15(0.62,2.1) | 1.12(.49, 2.66) | .747 |

Note: HH = household; AOR = adjusted odds ratio; COR = crude odds ratio, significant at
P-value<0.05.

*mobilization is the role of the health extension workers only. . .I have been trying my best to make the community a member of the scheme. . . .Some of our community are sparsely populated and pastoralist as well as hard to reach . . . we need strong managerial support. . .they have to work with us"* (In-depth interview, HEWs focal).

**Inconsistent CBHI implementation strategies and premium level.** Inconsistency in the premium level and CBHI implementation strategies between different districts were reported by most of the health extension workers.

*"I collected 355 birr per household . . .350 birr for regular contribution and 5 birr for registration. . .besides if there are18 years and above children, house maid, or other relatives in the house; I was collecting extra 75 birr. But other health extension workers in other districts collected different amount of money per household and collected different amounts of additional money for households having children 18 years and more, housemaid or other relatives. . ."*

*(In-depth interview, HEWs focal).*

**Fraud and abuse of the premium.** Some of the community based health insurance district coordinators and the health office head reported concern over fraud and abuse of the collected money, which may affect the trust of the community in the scheme management team.

But measures have been taken to stop further frauds. *"The health extension workers and the kebele administrators (who are responsible for collecting the premium in the study area) have been using the collected money for their own sake and some of them lend the money to others and some of them made a business using it."* (In-depth -interview: district CBHI-coordinator).

*Policy issues related to CBHI.* According to the Ethiopian health insurance agency, to start providing health service under CBHI at least 50% of eligible households in the district should be a member otherwise the service provision will not be started. This may pose a major challenge to start service provision with an available number of CBHI members.

*"One of the main bottleneck not to start CBHI service is government regulation . . .Because of 50% of the district eligible households are not member, we have not started providing the service for previously enrolled households. . .for me, the solution is that either to start in any number of households or making membership obligatory."(In-depth -interview: district health office head).*

## Discussion

The CBHI enrollment status in the current study was 33.3% [95% CI: 29.9–36.2]. In comparison, this figure was lower than the study conducted in Rwanda 85% [12], pilot districts of Ethiopia 52.4% [8], Northwest part of Ethiopia 42% [15], West Gojam Zone 58% [16] and Tanzania 49% [17]). This difference might be attributed to the level of influential community members (community leaders, religious leaders, elders, and health development team leaders) involvement in sensitization and awareness creation activities. This might have emanated from a lack of managerial commitment at the lower-level managers. Based on this finding we suggest community engagement in decision-making about the types of services, payment approach, and service delivery. Besides, continued political instability in the study areas might explain the lower enrollment. The result from the present study can be used as a signal for a need of much effort to attain the goal of the Ethiopian government health sector transformation plan to expand CBHI schemes to 80% of districts and enroll at least 80% of households by 2020 [18].

The current study showed that Knowledge regarding community-based health insurance affected the decision of enrollment positively. This finding was analogous to a study from another part of Ethiopia [19] and other African countries like Uganda, Kenya, and Nigeria; accordingly, limited information and poor knowledge limits voluntary enrollment and re-enrollment in the scheme [11, 20, 21]. The concept of insurance and risk pooling is relatively new for many people in low-income countries in general and remote areas like our study setting in particular. We can increase CBHI understanding and concept using different social marketing strategies including locally available means; and efficient information campaign as well as provision of training on the important parameters of CBHI, would contribute to the improvement of understanding and knowledge of the community on health insurance and definitely will increase the enrollment rate. Moreover, the educational status of the household head affected the decision of CBHI enrollment positively. This is also corroborated by the study finding of willingness to join CBHI in Ethiopia [22], enrollment studies in Kenya, India, and Bangladesh [20, 23, 24]. The finding can be explained as those educated household heads' are likely to have better acceptance and knowledge about the meaning as well as the benefit packages of CBHI and its protection from catastrophic out of pocket health expenditure at the time of ailment, which leads them to make a rational decision to enroll in the scheme. On the contrary, the study conducted in Tanzania showed no significant relation between education and CBHI enrollment [17].

In this study, households with larger family size had higher odds of CBHI enrollment than those with a smaller number of family members. The finding was in agreement with the study conducted in Ethiopia [15]. This can be justified as there was no variation with premium payment among different family size in Ethiopia except in Oromiya region; it is advantageous for households with larger family size to join the scheme. Household unit of enrollment can be an effective mechanism to address all family members. Besides, household enrollment decreased adverse selection due to a lower probability of having only sick and higher risk individuals enrolled in the scheme. In contrast, the three studies showed that households with larger family size are less likely to enroll in CBHI the reason might be due to there was a member restriction on the mentioned studies and they might face difficulties in meeting the subscription fees [11, 25, 26]. The present study also illustrates that household members with chronic disease, frequently ill individuals due to communicable disease in the household, and households' with under eighteen children and elders are more likely to enroll in the scheme than their counterparts. Even though, the scheme aimed to facilitate health service utilization and promote equitable distribution of health service among different segments of the population. Such practice may endanger risk-sharing principles, as those who have medical conditions and elders who have a probability of getting sick frequently involve actively, thus increase the chance of using all the resources within a short period exhaustively; and the result also shows us the existence of adverse selection problem. This evidence is also supported by the research done in the Northwest and Northeast part of Ethiopia [15, 27] and the two studies conducted in India [23, 28]. In this study, 44.1% of the participant reported that their main reason for not to enroll in CBHI scheme was poor quality of service provided in the public health facilities. This was also verified by FGD nonmember participants, "*The care given to us at the primary hospital/health center is poor. . ..there is no medication, even medication to treat headache in health centers/primary hospitals. . .. how we can be a member*?*".* This could reflect a negative attitude towards public facilities, meanwhile, the community have a good opinion towards CBHI, if that is the case, the government body would be better to fulfill the facilities and the recommended services should be provided for the community, so as more member will be enrolled in the CBHI scheme. This finding is also corroborated by the study conducted in Nigeria, Uganda, and Rwanda [6, 11, 29] members and nonmembers of CBHI schemes complained about the inconvenient facility like lack of drugs and supplies affect their enrollment. Moreover, in the other study conducted in Low- and Middle-Income Countries including Nigeria [6, 30] the result showed the perception of the community towards good quality healthcare provided in the public facilities or availability of quality service as a factor enhancing enrolment. In the qualitative finding of the present study, one of the CBHI coordinators and health extension workers complains that the main barrier not to enroll the community in the scheme was "*lack of managerial commitment at different level*". They also suggest the involvement of an academician in the development process of the CBHI scheme. This finding is also supported by the studies conducted in Uganda and Tanzania [11, 17], financial support from government and managerial commitment was reported to have a positive influence on CBHI enrollment and sustaining the scheme and sufficiently meeting the health needs of the communities. So, this results highlight lack of managerial commitment affects the CBHI enrollment. Moreover, our findings also show that there was lower involvement of the community on decision making regarding CBHI, fifty two percent of the participant reported that the community has no the right to guide and supervise the activities of the CBHI management. This result was also supported by FGD participants," *We have no idea how much birr is collected, how many households were enrolled. . .where the premium is pooled . . ."* This finding reflects the community-based ideology from which the CBHI was built is missing according to the responses from community members. It indicates a lack of solidarity and risk-sharing principles.

### Limitation of the study

The limitations of this study might be the cross-sectional nature of the data, which does not show cause and effect. This study depends on verbal responses and might therefore have included some misinformation due to recall bias. In dealing with this type of bias, we allowed participants to use reference materials and also arranged questions in order of providing an opportunity to recall. Also, there might be a social desirability bias. To minimize these limitations we have used different sources of information.

## Conclusion and recommendation

There was a lower enrollment status of the households in the community based health insurance (CBHI) scheme. We found that enrollment to community-based health insurance was higher among people who had good knowledge, higher family size, presence of children whose age ($\leq$18 years) and or elders (65+ years), presence of a household member with chronic non-communicable diseases, frequently ill individual due to communicable diseases and higher educational status of the households. Perceived poor quality of health care in terms of shortage of drugs and supplies, lack of trust of the community on the commitment of CBHI administrators, lack of transparency of CBHI scheme management were the reasons listed by the community for non-enrollment. Wide-range awareness creation activities through locally available social marketing strategies are beneficial. Capacity building for CBHI administrators and trust-building among communities through transparent management should be given attention. Besides, the quality of care being given in public health facilities should be improved. We suggest further operational research to assess the readiness of health facilities to accommodate the patients' expectations and the barriers in the study area.

## Supporting information

**S1 File. Pre-tested and structured questioners for factors affecting enrollment status of households for community based health insurance in a resource-limited peripheral area in Southern Ethiopia.**
(DOCX)

**S2 File.**
(SAV)

## Acknowledgments

First, we would like to thank all study participants, data collectors, supervisors, and study area Zonal Health Departments & District Health Offices. Second, we are very thankful for Arba Minch University, College of Medicine and Health Science staff for their constructive advice and support.

## Author Contributions

**Conceptualization:** Mustefa Glagn Abdilwohab, Zeleke Hailemariam Abebo.

**Data curation:** Mustefa Glagn Abdilwohab.

**Formal analysis:** Mustefa Glagn Abdilwohab.

**Funding acquisition:** Mustefa Glagn Abdilwohab.

**Investigation:** Mustefa Glagn Abdilwohab, Zeleke Hailemariam Abebo, Dessalegn Ajema, Manaye Yihune.

**Methodology:** Mustefa Glagn Abdilwohab, Zeleke Hailemariam Abebo, Wanzahun Godana, Dessalegn Ajema, Manaye Yihune, Hadiya Hassen.

**Project administration:** Mustefa Glagn Abdilwohab.

**Resources:** Mustefa Glagn Abdilwohab.

**Software:** Mustefa Glagn Abdilwohab, Zeleke Hailemariam Abebo, Dessalegn Ajema, Manaye Yihune.

**Supervision:** Mustefa Glagn Abdilwohab, Zeleke Hailemariam Abebo, Dessalegn Ajema, Manaye Yihune, Hadiya Hassen.

**Validation:** Mustefa Glagn Abdilwohab, Zeleke Hailemariam Abebo, Wanzahun Godana, Dessalegn Ajema, Hadiya Hassen.

**Visualization:** Mustefa Glagn Abdilwohab, Zeleke Hailemariam Abebo, Wanzahun Godana, Dessalegn Ajema, Manaye Yihune, Hadiya Hassen.

**Writing – original draft:** Mustefa Glagn Abdilwohab, Zeleke Hailemariam Abebo.

**Writing – review & editing:** Mustefa Glagn Abdilwohab, Zeleke Hailemariam Abebo, Wanzahun Godana, Dessalegn Ajema, Manaye Yihune, Hadiya Hassen.

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
