## [Decision Letter · Decision Letter 0]

4 Mar 2020

PONE-D-19-31045

Adverse selection and supply-side factors in the enrollment of community-based health insurance in peripheral zones of South Nation Nationalities People Region, Ethiopia:   Mixed Methodology

PLOS ONE

Dear Mr Abdilwohab,

Thank you for submitting your manuscript to PLOS ONE. After careful consideration, we feel that it has merit but does not fully meet PLOS ONE’s publication criteria as it currently stands. Therefore, we invite you to submit a revised version of the manuscript that addresses the points raised during the review process.

We would appreciate receiving your revised manuscript by Apr 18 2020 11:59PM. To enhance the reproducibility of your results, we recommend that if applicable you deposit your laboratory protocols in protocols.io, where a protocol can be assigned its own identifier (DOI) such that it can be cited independently in the future. For instructions see: http://journals.plos.org/plosone/s/submission-guidelines#loc-laboratory-protocols

We look forward to receiving your revised manuscript.

Kind regards,

Marta Pascual

Academic Editor

PLOS ONE

Journal Requirements:

2. Please address the following:

- Please modify the title to ensure that it is meeting PLOS’ guidelines (https://journals.plos.org/plosone/s/submission-guidelines#loc-title). In particular, the title should be "specific, descriptive, concise, and comprehensible to readers outside the field".  Please ensure that you amend the title both on the online submission form (via Edit Submission) and the title in the manuscript so that they are identical.

- Please ensure you have thoroughly discussed any potential limitations of this study within the Discussion section.

- Please include additional information regarding the survey or questionnaire used in the study and ensure that you have provided sufficient details that others could replicate the analyses. For instance, if you developed a questionnaire as part of this study and it is not under a copyright more restrictive than CC-BY, please include a copy, in both the original language and English, as Supporting Information.

Thank you for your attention to these queries.

4. We note you have included tables to which you do not refer in the text of your manuscript. Please ensure that you refer to Tables 2, 3 and 4 in your text; if accepted, production will need this reference to link the reader to each Table.

5. Please include captions for your Supporting Information files at the end of your manuscript, and update any in-text citations to match accordingly. Please see our Supporting Information guidelines for more information: http://journals.plos.org/plosone/s/supporting-information

Reviewers' comments:

Reviewer's Responses to Questions

**Comments to the Author**

1. Is the manuscript technically sound, and do the data support the conclusions?

Reviewer #1: Partly

Reviewer #2: No

Reviewer #3: Partly

Reviewer #4: Yes

2. Has the statistical analysis been performed appropriately and rigorously? 

Reviewer #1: No

Reviewer #2: I Don't Know

Reviewer #3: Yes

Reviewer #4: Yes

3. Have the authors made all data underlying the findings in their manuscript fully available?

Reviewer #1: No

Reviewer #2: Yes

Reviewer #3: Yes

Reviewer #4: Yes

4. Is the manuscript presented in an intelligible fashion and written in standard English?

Reviewer #1: No

Reviewer #2: No

Reviewer #3: Yes

Reviewer #4: No

5. Review Comments to the Author

Reviewer #1: The manuscript by Abdilwohab et al., attempted to identify and access factors affecting enrollment status of households for community-based health insurance in a resource-limited peripheral area in South Ethiopia. A relatively new way of some governments in Africa to provide universal health coverage to low-income people by sharing the cost. However, I have multiple issues with the manuscript:

1. The study presents the results of original research.

• The manuscript presents data from a study that was conducted among selected 820 Households from April 27 to June 12, 2018, in a rural area of Southern Ethiopia.

• Although in my opinion, there is nothing new in regards to the design, the aims or the results of this study because of its similarity with multiple published articles. More particularly a recent published paper with almost word by word title from Northwest Ethiopia area (Atafu et al.,2018, Int J Health Plann Mgmt). However, this study could probably still benefit this particular community.

• The authors should consider editing the title to make it more comprehensive and create enough variability with the above-mentioned manuscript

• The authors should be careful of plagiarism in their abstract conclusion because of the fact that this reading almost similar to the paper referred to above.

2. Results reported have not been published elsewhere.

• Multiple recent publications have addressed this topic in similar communities of interest, and this manuscript could benefit from the literature, but the authors either did not review the literature or failed to acknowledge previous work.

• Examples: Mirach et al, 2019: "Determinants of community-based health insurance implementation in west Gojjam zone, Northwest Ethiopia: a community-based cross-sectional study design" and Bodhisane et al., 2018: "Factors affecting the willingness to join community‐based health insurance (CBHI) scheme: A case study survey from Savannakhet Province, Lao P.D.R".

• The study aim(s) and the recommendations from the manuscript by Abdilwohab et al., are obvious and have been the same for most of the above studies despite the fact that thy are are performed in different community

3. Experiments, statistics, and other analyses are performed to a high technical standard and are described in sufficient detail.

• The experiment design, statistics, and other analyses that were performed in this study are theoretically justified. In fact, the authors should be commended for they afford to calculate the cohort size and the size enrolled participants. However, there are multiple issues with the statistics and data presentation.

• The strength of this study should be the fact that authors combined a quantitative multivariate logistic regression and a qualitative population survey approach. Unfortunately, this does not appear to benefit the results, rather great a confusion. The structure should be normally that the authors will use the quantitative data to make an argument and support that with one or two testimonials from the interviews (testimonial). Wherever the authors don't have quantitative data to support a point, they should use one or two testimonials and state at least how many participants gave a similar point of view in they testimonials to illustrate the point. Unfortunately, in this study, the quantitative and qualitative data are not in coherent manner that support the study conclusion.

• The authors had access to enough statistical tools but they did not take advantage of them (Epi, SPSS,...). Normally, the qualitative data should also be binary coded and quantitatively analyzed using methods such as chi-square to calculate statistical significance.

• The authors referred to a principal comment analysis (PCA), but the results of such analysis are not shown anyway in the manuscript. Also, when using relatively unfamiliar data analysis methods such as PCA for basic analysis, the authors should give a rationale.

• The authors correctly focus on logistic regression analysis for most part of the data analysis. However, they still some confusion. The authors state that all variables from univariable regression with a p-value less or equal to 2 were put back into a multivariable regression, but there are no p-values shown anywhere in the tables or in the body of this manuscript.

• On page 12, the authors described about 12 variables that were returned into the multivariable analysis model, but "Table 4" (which supposed to be Table 5), only show 7 variables.

• The tables are miss-numbered (for example we have two 'table 2s', and so on), contain a tremendous amount of structural, grammatical and spelling errors that distract the reader from following. These tables could benefit as traditionally done; making the description of the variables shorter in the table, and then use of a table legend to describe the abbreviations.

• Table 1 need some structural and grammatical editing.

• The data in Table 2 is contradictory form the author's claim in the text. Row# 3 and 4 in table 2 demonstrate that if you combine the percentages of participants who answered incorrectly and those who did not know, that numbers is higher than those who answer correctly, suggesting that in general participants did not understand the concept or did not know how the CBHI system work. However, in the paragraph below the table, the authors only use the percentage from the “correct” column to conclude that participants have a good knowledge and understanding of CBHI.

4. Conclusions are presented in an appropriate fashion and are supported by the data.

• The conclusions are supported by the data but are obvious and similar or exactly the same to most recent publications that the authors either are not way of or failed to acknowledge. It will beneficial to the field if the authors relate their results to these most recent publications.

• The authors failed to make an important point that keeps appearing from the testimonials of participants. It seems like the CBHI administrators have failed to manage the expectation of the community. In some testimonials, participants are admitting that the premium are very affordable but the public hospitals/clinics services are very bad. The simple solution could be to educate the community that they can only be able to get the services that the CBHI is able to pay for regarding the low premium. It will be beneficial to the recommendations if the authors speculate on that point.

5. The article is presented in an intelligible fashion and is written in standard English.

• It is understandable that the authors are not English-first language speakers, but the manuscript has a significant structural, grammatical and spelling errors that distract the reader. The manuscript could highly benefit from an English editor.

• For example, the last paragraph of the abstract background is structurally and grammatically distractive and it could even benefit from basic online English editing tools (eg. Grammarly)

6. The research meets all applicable standards for the ethics of experimentation and research integrity.

• The authors state that the Ethical approval was obtained.

7. The article adheres to appropriate reporting guidelines and community standards for data availability.

• The authors must be commended for the number of interviews and the number of questionnaires they conducted during this study. One assumes all the records are available.

Reviewer #2: The authors investigated supply-side and adverse selection factors in the enrollment in community-based health insurance (CBHI) in selected zones in Ethiopia, Africa. The research topic/area is of interest towards providing good health care coverage to the sampled citizens. However, the manuscript cannot be accepted as it currently stands because of its quality and a number of issues which should be addressed by the authors. I recommend a Major review and re-submission subject to the corrections of the under-listed issues.

(1) The title of the manuscript is too long and clumsy. Authors should among other titles, consider: Determining the factors influencing the selection and enrollment in community-based health insurance scheme in Ethiopia.

(2) The study appears to lack novelty and have few innovations that could be readily pointed to. No new method(s)/technique(s) was introduced as authors just applied existing method(s) to their study without any creativity attached.

Authors should specifically mention their study's innovations and contributions that makes it unique and state-of-the-art.

The need for utilizing some methods including thematic analysis, narrative weaving, binary logistic regression etc should be briefly explained.

(3) It is suggested that authors should include the sample questionnaires administered in the study as an addendum to the manuscript to enable the reviewers and other researchers have a better understanding.

(4) The manuscript is not well structured and written in standard English language. There exist a lot of typos and grammatical errors that must be resolved. These affects the readability and understanding of the manuscript. For instance:

(i) inappropriate mixing, use and introduction of various upper case letters

(ii) poor use and outright omission of punctuation marks. Check the list of Authors' name, Authors' affiliations, Abstract

including wrong indexing (superscripts)

(iii) heath - instead of health

(iv) "The study instruments/tools for this study..."

(v) "health development army"

(vi) "The other reason for low reason..."

(5) Woreda; kebele: First appearance of these words on page 2 and 3 respectively does not indicate their meaning.

(6) What informed the authors' selection of the Segen area and South Omo Zones for this research study especially as it was also stated in the Discussion section that these areas are prone to political instability?

(7) Some of the statements in the manuscript are vague. For instance on page 3,

(i) "The administrative offices of the zones are located about 750kms south of Addis Ababa".

Authors should state what administrative offices are.

(ii) "But one of the district's population are semi-pastoralist and pastoralist" - which district?

(8) There are two Table 2 in the manuscript - kindly resolve.

Authors should also stick to just one of these words "Not Correct" or "Incorrect" as shown in Table 2 to aid understanding.

(9) The various Tables in the manuscript are not explained, referred to and properly linked to the statements in the body of the manuscript. Some of these Tables are simply counting, percentages or frequency related and not statistically rich enough. Some acronyms e.g. HH in Table 2 are not provided meanings.

(10) Consider revising the captions of the Tables in the manuscript. They are too long and contains repeated phrases.

(11) The factors identified from the qualitative results and listed on page 14 were just presented as obtained from respondents without authors' input and explanations including their impacts and relating these findings to the current study.

(12) Numbered references in the body of the manuscript should be in square brackets. Some references doesn't even have yaer of publication stated. The referencing style adopted by the authors is not consistent. Kindly check and adapt to Plosone reference style.

Reviewer #3: The manuscript is technically sound and the methodology was done with rigor. The study investigates a critical aspect of access to health care that can significantly contribute to UHC. My main comment is the conclusion appears to be biased towards identifying adverse selection factors such as age, illness, education etc. On the other hand majority of non CBHI users indicated that lack of quality of service is the reason why they are not enrolled in the CBHI. I recommend the availability of quality services to be adequately reflected in the conclusion.

Reviewer #4: Universal health coverage is essential to ensure global access to health care. It is an important subject for research in resource poor settings. This study uses sound methodology to investigate possible factors affecting the uptake of CBHI. The authors have made a number of typographical and grammatical errors. They use terms not of English origin and have omitted giving meaning to these words. The discussion needs more work in drawing comparisons with other similar research to draw conclusions, make adequate recommendations and suggestions for further research. I have uploaded a copy of the manuscript with comments.

6. PLOS authors have the option to publish the peer review history of their article (what does this mean?). If published, this will include your full peer review and any attached files.

Reviewer #1: Yes: Ngomu Akeem Akilimali

Reviewer #2: No

Reviewer #3: Yes: Muna Abdullah Ali

Reviewer #4: Yes: Febisola Ajudua

---

## [Author Response · Author response to Decision Letter 0]

2 Apr 2020

The authors response for the questions raised by the reviewers is uploaded as 'response to reviewers'

---

## [Decision Letter · Decision Letter 1]

12 Nov 2020

PONE-D-19-31045R1

Factors affecting enrollment status of households for community based health insurance in a resource-limited peripheral area in Southern Ethiopia.    Mixed method

PLOS ONE

Dear Dr. Abdilwohab,

Thank you for submitting your manuscript to PLOS ONE. After careful consideration, we feel that it has merit but does not fully meet PLOS ONE’s publication criteria as it currently stands. Therefore, we invite you to submit a revised version of the manuscript that addresses the points raised during the review process.

Please ensure that you copyedit your manuscript for English usage and grammar. In addition, please address those reporting issues highlighted by reviewer #1, particularly regarding table 2 and the result interpretation.

We look forward to receiving your revised manuscript.

Kind regards,

Sara Fuentes Perez

Staff editor,

PLOS ONE

On behalf of:

Joseph Telfair, DrPH, MSW, MPH

Academic Editor

PLOS ONE

Reviewers' comments:

Reviewer's Responses to Questions

**Comments to the Author**

1. If the authors have adequately addressed your comments raised in a previous round of review and you feel that this manuscript is now acceptable for publication, you may indicate that here to bypass the “Comments to the Author” section, enter your conflict of interest statement in the “Confidential to Editor” section, and submit your "Accept" recommendation.

Reviewer #1: (No Response)

Reviewer #3: All comments have been addressed

2. Is the manuscript technically sound, and do the data support the conclusions?

Reviewer #1: Partly

Reviewer #3: Yes

3. Has the statistical analysis been performed appropriately and rigorously? 

Reviewer #1: I Don't Know

Reviewer #3: Yes

4. Have the authors made all data underlying the findings in their manuscript fully available?

Reviewer #1: Yes

Reviewer #3: Yes

5. Is the manuscript presented in an intelligible fashion and written in standard English?

Reviewer #1: No

Reviewer #3: Yes

6. Review Comments to the Author

Reviewer #1: 2. Is the manuscript technically sound, and do the data support the conclusions?

The theoretical analysis is technically sound but is not clear if the authors understand the results and maybe misinterpreting the results. Just a few examples after revision:

- Table 2 with the title "Table 2: Knowledge of the study participants related to CBHI in Segen area and South Omo zones, Southern, Ethiopia 27April to 12 June, 2018. If you calculate the average 'correct response' vs 'incorrect response' based on 5 knowledge questions in that Table 2 is 426.4 vs 393.6 participants. However, line239-241 the authors write: A total of 446 (54.4%) of the study participants had poor knowledge related to community based health insurance and the remaining 45.6% of the participants had good knowledge related to CBHI. ‘’Table 2’’

- line249: One may assume that 5 knowledge questions in Table 2 were the top rank reasons in the knowledges-base analysis therefore the author decided to focus on those but is not clear in the manuscript.

3.Has the statistical analysis been performed appropriately and rigorously?

A couple of statistical analysises such as PCA, thematic analysis method,...are referred to by the author put there is no explanation of these analyses or proper presentation of results from these analyses. The author failed to appropriately attend those reviewer's requests.

5.Is the manuscript presented in an intelligible fashion and written in standard English?

There are a lot of English related issues that the authors failed to attend too after editing:

-Inappropriate mixing and alternation of use low/upper case

-Poor use and outright omission of punctuation marks.

-A lot of grammar, and spelling errors...

Reviewer #3: The author has adequately addressed the comments provided. I am of the opinion that the study will contribute to the much needed discussion around universal health coverage and effective health coverage.

7. PLOS authors have the option to publish the peer review history of their article (what does this mean?). If published, this will include your full peer review and any attached files.

Reviewer #1: **Yes: **Ngomu Akeem Akilimali

Reviewer #3: **Yes: **Muna Abdullah Ali

---

## [Author Response · Author response to Decision Letter 1]

29 Nov 2020

Response to reviewer comments

6. Review Comments to the Author

Reviewer #1: 2. Is the manuscript technically sound, and do the data support the conclusions?

The theoretical analysis is technically sound but is not clear if the authors understand the results and maybe misinterpreting the results. Just a few examples after revision:

- Table 2 with the title "Table 2: Knowledge of the study participants related to CBHI in Segen area and South Omo zones, Southern, Ethiopia 27April to 12 June, 2018. If you calculate the average 'correct response' vs 'incorrect response' based on 5 knowledge questions in that Table 2 is 426.4 vs 393.6 participants. However, line239-241 the authors write: A total of 446 (54.4%) of the study participants had poor knowledge related to community based health insurance and the remaining 45.6% of the participants had good knowledge related to CBHI. ‘’Table 2’’

- line249: One may assume that 5 knowledge questions in Table 2 were the top rank reasons in the knowledges-base analysis therefore the author decided to focus on those but is not clear in the manuscript.

Response: Dear reviewer, if we are not mistaken your concern is related to the measurement of knowledge. As you know knowledge is a variable that is impossible to measure using one item/variable. For this reason, we have adopted a measurement tool from Ethiopian health insurance agency prepared to measure the knowledge of CBHI members in the country. Based on the response of the study participants in the five items we have calculated the knowledge of the study participants as described below.

Participants’ Knowledge about CBHI was assessed using five questions that consisted of concepts, roles, and beneficiaries’ of community-based health insurance. Participants correctly responded to these questions were categorized as ‘correct response’ otherwise ‘not correct response’. Each item was equally weighted. Thus, each correct response had a score of 1 and each wrong response had a score of 0. Hence, the aggregate score for all knowledge questions would range from 0–5 points. Participants’ overall knowledge was categorized as good if the score was 4 and 5 (>= 70%) points and otherwise poor. Kindly see in the revised manuscript method section line 134- 140.

CBHI members who scored greater than or equal to 70 percent of the sum score of all items were considered as having good knowledge. This threshold is recommended by experts in the field of study and the Agency too. We haven’t calculated the average. Hope that you will understand the way we measured the knowledge of CBHI members mathematically. Thank you very much for raising your concern. 

3.Has the statistical analysis been performed appropriately and rigorously?

A couple of statistical analysises such as PCA, thematic analysis method,...are referred to by the author put there is no explanation of these analyses or proper presentation of results from these analyses. The author failed to appropriately attend those reviewer's requests.

Response: for more clarification, we have described principal component analysis (PCA) in the method section under subsection “principal component analysis” kindly look at in the revised manuscript line 190-203.

The result of the Principal component is described in Table 1 as “Wealth quantile” and it is highlighted in the red color in the “Revised manuscript with track changes”. Kindly look at Table 1 stated as wealth quantile in the revised manuscript as well.

Thematic analysis is a method for identifying, analyzing, and reporting patterns (themes) within data. It minimally organizes and describes your data in (rich) detail. 

 Ref Braun & Clarke, 2006:79

Thematic analysis is the commonest method of analysis of qualitative data. Experts in the area of public health used this method to group related quotes together in a qualitative study. We have explained well how we have thematized the quotes and/or codes in the method section under subsection “Qualitative data analysis” line 204- 216. 

In the result section: In the result section, we have described each quote under each theme based on their relationship in meaning and concepts. Here are some of the themes that emerged during the analysis of our qualitative data: Perceived quality of care, Trust management, Community involvement, Lack of managerial commitment, Inconsistent CBHI implementation strategies and premium level, Fraud and abuse of the premium, and Policy issues related to CBHI. 

 Kindly see the result section in the revised manuscript. 

This is how we explained PCA and thematic analysis in the method section and how we presented the result in the result section. Hope that the reviewer will satisfy with our explanation.

Thank you again for sharing your concern

5.Is the manuscript presented in an intelligible fashion and written in standard English?

There are a lot of English related issues that the authors failed to attend too after editing:

-Inappropriate mixing and alternation of use low/upper case

-Poor use and outright omission of punctuation marks.

-A lot of grammar, and spelling errors...

Response: Thank you so much for your meticulous observation. 

We have revised the document for English editing. Kindly see the revised manuscript.

Thank you so much for reviewing our manuscript and sharing your professional experience!

---

## [Decision Letter · Decision Letter 2]

24 Dec 2020

PONE-D-19-31045R2

Factors affecting enrollment status of households for community based health insurance in a resource-limited peripheral area in Southern Ethiopia.    Mixed method

PLOS ONE

Dear Dr. Abdilwohab,

Thank you for submitting your manuscript to PLOS ONE. After careful consideration, we feel that it has merit but does not fully meet PLOS ONE’s publication criteria as it currently stands. Therefore, we invite you to submit a revised version of the manuscript that addresses the points raised during the review process.

ACADEMIC EDITOR comment.

There was challenge getting qualified reviewers for this manuscript, the challenge has been met.

The authors are strongly encouraged to attend to the comments and re-submit of the reviewers if the manuscript is to be considered for publication.

We look forward to receiving your revised manuscript.

Kind regards,

Joseph Telfair, DrPH, MSW, MPH

Academic Editor

PLOS ONE

Reviewers' comments:

Reviewer's Responses to Questions

**Comments to the Author**

1. If the authors have adequately addressed your comments raised in a previous round of review and you feel that this manuscript is now acceptable for publication, you may indicate that here to bypass the “Comments to the Author” section, enter your conflict of interest statement in the “Confidential to Editor” section, and submit your "Accept" recommendation.

Reviewer #5: All comments have been addressed

Reviewer #6: (No Response)

2. Is the manuscript technically sound, and do the data support the conclusions?

Reviewer #5: Partly

Reviewer #6: Yes

3. Has the statistical analysis been performed appropriately and rigorously? 

Reviewer #5: Yes

Reviewer #6: Yes

4. Have the authors made all data underlying the findings in their manuscript fully available?

Reviewer #5: No

Reviewer #6: Yes

5. Is the manuscript presented in an intelligible fashion and written in standard English?

Reviewer #5: Yes

Reviewer #6: No

6. Review Comments to the Author

Reviewer #5: (No Response)

Reviewer #6: There are still grammatical errors that the authors did not attend to:

line 30: " and identifying factors affecting it in peripheral areas of Southern Ethiopia" should read " and identify factors affecting it in peripheral areas of Southern Ethiopia".

168-169: No punctuation mark

241, 249: Tenses should be in the past

Authors should read the manuscript and correct these grammatical blunders.

Abbreviations used in text should be explained, line 286

7. PLOS authors have the option to publish the peer review history of their article (what does this mean?). If published, this will include your full peer review and any attached files.

Reviewer #5: No

Reviewer #6: No

---

## [Author Response · Author response to Decision Letter 2]

28 Dec 2020

To: PLOS ONE editorial office

Subject: Response to reviewer comments

This is a point-by-point response to the reviewer comments we received as a second revision for our article entitled “Factors affecting enrollment status of households for community-based health insurance in a resource-limited peripheral area in Southern Ethiopia. Mixed method” -

PONE-D-19-31045R1. We have uploaded the revised manuscript both clean and track changes

Versions. Kindly look at the revisions and rebuttals we have made in line with each review point.

Hope to hear from you shortly.

Kind regards,

Mustefa Glagn (corresponding author)

Email:mustesami02@gmail.com

Response to Reviewers

2. Is the manuscript technically sound, and do the data support the conclusions?

Reviewer #5: Partly

Response: as far as our knowledge is concerned, when we conclude the research findings we need to summarize our thoughts, highlight key points in our analysis or findings, and contextualizing the research problem based on the results of our study. Besides, we need to answer our research questions. The following were our research questions:

1. What proportion of households was enrolled in community-based health insurance in a resource-limited peripheral area in Southern Ethiopia?

2. What are the factors affecting the enrollment status of households for community-based health insurance in a resource-limited peripheral area in Southern Ethiopia?

We have tried to address our research questions in our conclusion. The study found that lower community-based health insurance enrollment status. This statement describes the first research questions. We said low enrollment because the Ethiopian government expects 100% CBHI enrollment in the study area in the given period but our research findings showed only 33.30% of households were enrolled in the CBHI. 

A higher probability of CBHI enrollment among higher health care demanding population groups was observed. Poor perceived quality of health care, poor managerial support, and lack of trust were found to be barriers for non-enrollment. This statement describes the factors which affect CBHI enrollment. These factors are identified in our research. So this addresses the second research question.

In the conclusion section, we also need to recommend a specific course or courses of action based on our findings. Here is our recommendation based on our findings: Therefore, wide-range awareness creation strategies should be used to address adverse selection and poor knowledge. In addition, trust should be built among communities through transparent management. Furthermore, the quality of care being given in public health facilities should be improved to encourage the community to be enrolled in CBHI. 

Based on our evaluation we have drawn the conclusion from our research findings and we have tried to address all the components needed to be addressed in the conclusion section.

Hope that you will be satisfied with our explanation.

Reviewer #6: Yes

4. Have the authors made all data underlying the findings in their manuscript fully available?

Reviewer #5: No

Response: All relevant data are within the manuscript and Supporting Information files. We have attached the raw data supporting the findings of this research and we also attached the questionnaire. 

Reviewer #6: Yes

6. Review Comments to the Author

Reviewer #5: (No Response)

Reviewer #6: 

There are still grammatical errors that the authors did not attend to:

line 30: " and identifying factors affecting it in peripheral areas of Southern Ethiopia" should read " and identify factors affecting it in peripheral areas of Southern Ethiopia".

168-169: No punctuation mark

241, 249: Tenses should be in the past

Authors should read the manuscript and correct these grammatical blunders.

Abbreviations used in text should be explained, line 286

Response: Thank you very much for your meticulous observation of the manuscript. We have corrected in the revised manuscript.

 Thank you all!

 Appreciated!

---

## [Decision Letter · Decision Letter 3]

12 Jan 2021

Factors affecting enrollment status of households for community based health insurance in a resource-limited peripheral area in Southern Ethiopia.    Mixed method

PONE-D-19-31045R3

Dear Dr. Abdilwohab,

We’re pleased to inform you that your manuscript has been judged scientifically suitable for publication and will be formally accepted for publication once it meets all outstanding technical requirements.

Kind regards,

Joseph Telfair, DrPH, MSW, MPH

Academic Editor

PLOS ONE

Additional Editor Comments (optional):

Reviewers' comments:

Reviewer's Responses to Questions

**Comments to the Author**

1. If the authors have adequately addressed your comments raised in a previous round of review and you feel that this manuscript is now acceptable for publication, you may indicate that here to bypass the “Comments to the Author” section, enter your conflict of interest statement in the “Confidential to Editor” section, and submit your "Accept" recommendation.

Reviewer #5: All comments have been addressed

Reviewer #6: All comments have been addressed

2. Is the manuscript technically sound, and do the data support the conclusions?

Reviewer #5: Yes

Reviewer #6: Yes

3. Has the statistical analysis been performed appropriately and rigorously? 

Reviewer #5: Yes

Reviewer #6: Yes

4. Have the authors made all data underlying the findings in their manuscript fully available?

Reviewer #5: No

Reviewer #6: Yes

5. Is the manuscript presented in an intelligible fashion and written in standard English?

Reviewer #5: Yes

Reviewer #6: Yes

6. Review Comments to the Author

Reviewer #5: (No Response)

Reviewer #6: (No Response)

7. PLOS authors have the option to publish the peer review history of their article (what does this mean?). If published, this will include your full peer review and any attached files.

Reviewer #5: No

Reviewer #6: No

---

## [Editor Report · Acceptance letter]

14 Jan 2021

PONE-D-19-31045R3 

Factors affecting enrollment status of households for community based health insurance in a resource-limited peripheral area in Southern Ethiopia.    Mixed method      

Dear Dr. Abdilwohab:

I'm pleased to inform you that your manuscript has been deemed suitable for publication in PLOS ONE. Congratulations! Your manuscript is now with our production department. 

Kind regards, 

on behalf of

Dr. Joseph Telfair 

Academic Editor

PLOS ONE